# Physiological and Genetic Analysis of Leaves from the Resprouters of an Old *Ginkgo biloba* Tree

**Jiali Yan †, Sixuan Zhang †, Miaomiao Tong, Jinkai Lu, Tongfei Wang, Yuan Xu, Weixing Li** 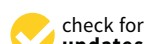 **and Li Wang ***

College of Horticulture and Plant Protection, Yangzhou University, Yangzhou 225009, China; 181803124@yzu.edu.cn (J.Y.); 181803133@yzu.edu.cn (S.Z.); 181803120@yzu.edu.cn (M.T.); dx120200123@yzu.edu.cn (J.L.); W173238040@163.com (T.W.); xuyuanyzu@163.com (Y.X.); liwx@yzu.edu.cn (W.L.)
* Correspondence: liwang@yzu.edu.cn; Tel.: +86-514-87979344; Fax: +86-514-87347537
† These authors contributed equally to this work.

**Abstract:** *Ginkgo biloba* is a well-known long-lived tree with important economical, ornamental and research value. New stems often resprout naturally from the trunk or roots of old trees to realize rejuvenation. However, the physiological and molecular mechanisms that underlie the resprouting from old trees are still unknown. In this study, we investigated a 544-year-old female ginkgo tree with vigorous resprouters along the trunk base in Yangzhou, China. We compared the morphological and physiological traits of leaves between resprouters (SL) and old branches (OL) and found a significantly higher thickness, fresh weight, and water content in SL. In particular, the depth and number of leaf lobes were dramatically increased in SL, suggesting the juvenile characteristics of sprouters in old ginkgo trees. Transcriptome data showed that the expression of genes related to photosynthetic capacity, the auxin signaling pathway, and stress-associated hormones was upregulated in SL. Importantly, levels of the most important secondary metabolites, including kaempferol, isorhamnetin, ginkgolide A, ginkgolide B, and ginkgolide C, were significantly higher in SL. We also identified high expression of key genes in SL, such as *PAL* and *FLS*, which are involved in flavonoid synthesis, and *GGPS*, which is involved in the synthesis of terpene lactones. These findings reveal the distinct physiological and molecular characteristics as well as secondary metabolite synthesis in leaves of resprouting stems in old ginkgo trees, providing new insight into rejuvenation physiology in old tree aging.

**Keywords:** *Ginkgo biloba*; old tree; resprouting; leaves; flavonoids; terpene lactones; RNA sequencing

## 1. Introduction

Plant development generally progresses through the following distinct phases: vegetative growth, reproductive phase, seed set, and senescence [1]. These phases are conspicuous in perennial species. Plant development differs from that in animals in that trees can undergo a reverse developmental process [2]. Rejuvenation refers to the disappearance of mature or semi-mature organs, tissues, and cells, as well as the recurrence of juvenile characteristics [3]. Resprouting is a form of plant self-repair and a common phenomenon in woody plants. When a tree is pruned, the injured branches can regenerate adventitious buds and develop new branches [4–8]. Therefore, resprouting is a critical mechanism by which tree renewal is maintained [9,10]. Resprouting can be classified into axillary, epicormic branch and stem, and basal types [11]. Among them, basal resprouting retain the characteristics of their parent tree and, given that the resprouting process is a natural part of tree development, comprise ideal material for rejuvenation.

*Ginkgo biloba* L., a fascinating and economically important gymnosperm, is the only living representative of the order Ginkgoales, which probably originated circa 280 million years ago [12]. It exhibits some unique features related to its sexual reproduction process, phylogenetic position, and evolutionary history [13–15]. At present, *G. biloba* has a wide

distribution across different elevations and latitudes and generally has a long lifespan. Indeed, hundreds of *G. biloba* trees aged over 1000 years still grow well and luxuriantly in China [16]. Although some studies have found that gymnosperms rarely undergo renewal via resprouters [17,18], resprouting often develops at the base of old ginkgo trees [19]. In naturally growing old ginkgo trees, new stems often develop at the base of the trunk, especially when the terminal buds of branches were destroyed or as the tree body ages. These new stems, also known as resprouters, originate from fixed latent buds above the stem/root transition zone. Generally, resprouters are thick and erect, which makes the position effect disappear obviously, and they are well suited for use in ginkgo cultivation and production [19].

*G. biloba* exhibits considerable environmental adaptability and resistance to pathogens. The latter trait is attributed to the presence of abundant secondary metabolites, including flavonoids and ginkgolides, in its leaves [20,21]. However, previous studies have revealed that in ginkgo, the leaves of young seedlings are larger, with more leaf lobes, and contain higher levels of flavonoids and ginkgolides. Conversely, as the tree develops, the leaf area becomes smaller, the number of leaf lobes and levels of medicinally active components decrease [22]. These findings imply that leaf morphology and the accumulation of medicinally active components may be closely related to age in ginkgo.

In this study, the morphology, physiology, and accumulation of secondary metabolites were compared between leaves from resprouters and those from old branches. To this end, we sampled leaves from a 544-year-old ginkgo tree growing on basal resprouters and old branches. Differences in morphology were assessed, physiological indices were measured, and RNA sequencing (RNA-seq) was performed in leaf samples. The purpose of our study is to investigate how the old ginkgo tree rejuvenates from resprouters by comparing the physiological and molecular characteristics of leaves between resprouters and old branches. Our results provide an important reference for studying the rejuvenation of old trees.

## 2. Materials and Methods

### 2.1. Plant Materials

In June 2020, we collected leaf samples from the old branches (OL) (Figure 1A–C) and resprouting stems (SL) (Figure 1D,E) of a 544-year-old (Yangzhou Ancient and Famous Trees Compilation) female *G. biloba* tree growing under natural conditions at Yangzhou of Jiangsu Province (32°39′ N, 119°43′ E), China (Figure 1A). For old branch sampling, we selected the leaves from the lateral branches in the southern periphery of outer middle crown. For resprouter sampling, we collected leaves from the resprouting stems at the south base trunk. All samples were immediately frozen in liquid nitrogen after collection and stored at −80 °C until use.

### 2.2. Measurement of Leaf Morphology and Weight

The leaves were photographed and then outlined in red in the resulting images using Image J software. A scale was set up to calculate the leaf area in each image. The thicknesses of 10 leaves were measured using a vernier caliper, and the number of leaf lobes per leaf was counted. These measurements were repeated three times.

To measure leaf fresh weight, 15 leaves were randomly selected from the SL and OL groups in each biological replicate; three biological replicates were set. The leaves were placed in an oven at 75 °C to dry them to a constant weight for dry weight measurements. The leaf water content was calculated as (fresh weight-dry weight)/fresh weight.

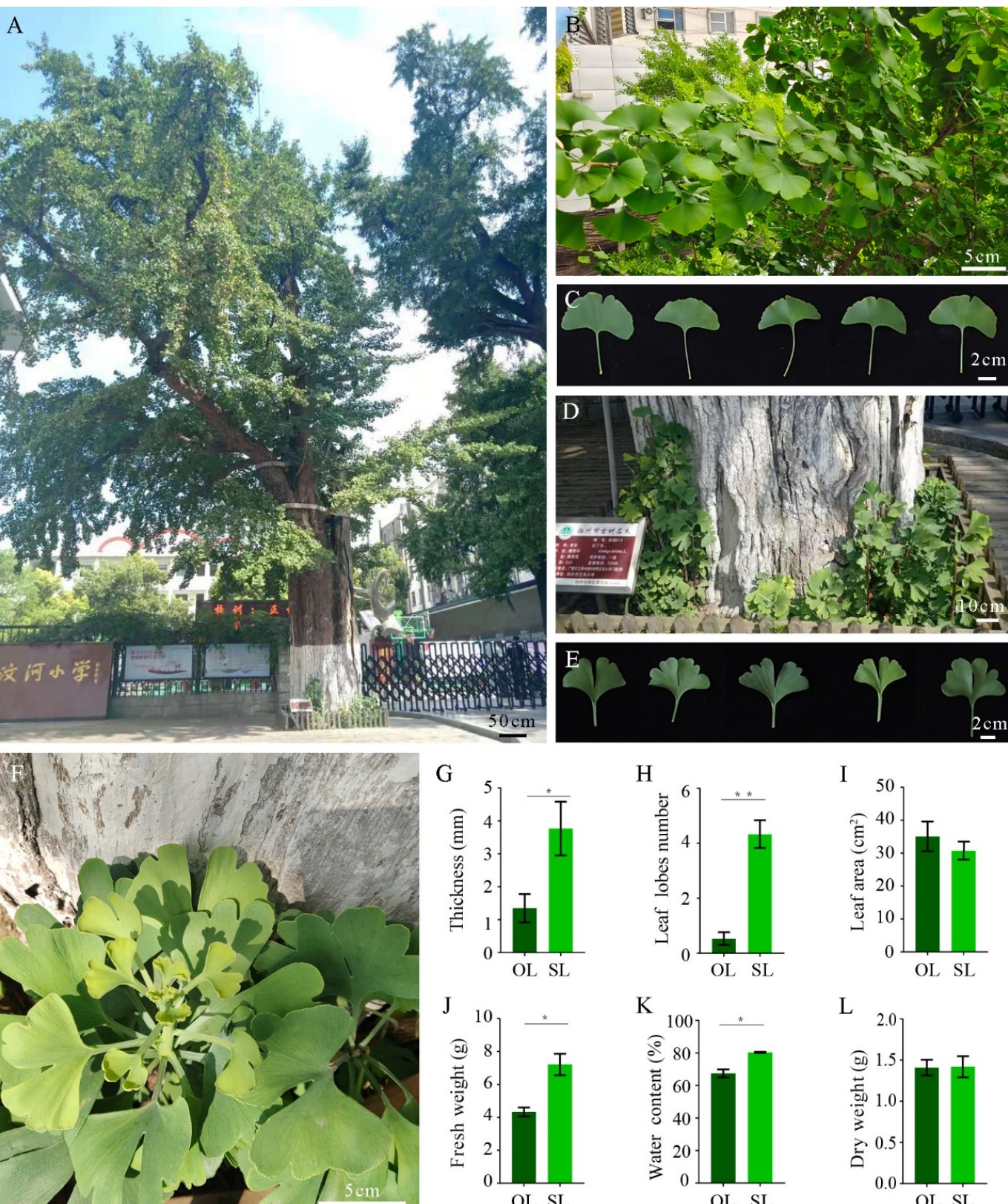

**Figure 1.** (**A**) The old female tree sampled in this study; (**B**) Branches in old tree; (**C**) Representative OL samples. Bar = 2 cm; (**D**) Sprouts at the tree base; (**E**) Representative SL from (**F**) leaves growing on sprouts. Bar = 2 cm; Leaf (**G**) thickness; (**H**) lobe number; (**I**) area; (**J**) fresh weight; (**K**) water content; (**L**) dry weight. Error bars represent the mean $\pm$ SD ($n = 3$). Asterisks above the columns indicate significant differences based on one-way ANOVA (* $p < 0.05$, ** $p < 0.01$).

### 2.3. Extraction and Quantification of Flavonol and Terpenoid Constituents

The flavonol and terpenoid contents in the leaves were measured by high-performance liquid chromatography (HPLC)-mass spectrometry (MS). For flavonol extraction, 1 g leaves were dried in an oven at 75 °C and then pulverized into powder. Analysis of the flavonol contents was performed following the method of the Chinese Pharmacopoeia Commission. Liquid chromatography (LC)-MS analysis was performed as described previously [23], using an LC-MS system (Agilent 6460; Agilent Technologies) with a C18 column (150 × 2.1 mm, 5 μm). Flavonol contents were analyzed based on ultraviolet-visible absorption at 370 nm. MS analysis of flavonols was performed following the methods described in Zhao et al. (2020). In addition, the total flavonoid content was calculated by multiplying total quercetin, kaempferol, and isorhamnetin by a factor of 2.51 [23].

An Atlantis T3 column (4.6 mm × 150 mm, 4 μm; Waters) was used to separate terpene lactones, which were analyzed using the Waters 2424 HPLC system. The mobile phase consisted of tetrahydrofuran, methanol, and water at a 10:25:65 (*v/v/v*) ratio. The elution profile was set as follows: flow rate of tetrahydrofuran–methanol, 0.35 mL/min; flow rate of water, 0.65 mL/min; column temperature, 35 °C. The total content (mg/g) of terpene lactones was determined as the sum of the contents of ginkgolide A, ginkgolide B, ginkgolide C, and bilobalide.

### 2.4. Transcriptome Sequencing

OL and SL were collected for RNA extraction. Six libraries (SL and OL groups) were sequenced on the Illumina Novaseq platform. Clean reads were obtained by removing adaptors and low-quality sequences with unknown nucleotides. At the same time, the Q20, Q30, and GC content of the clean reads were calculated. All downstream analyses were performed using clean reads of high quality. We performed de novo assembly of the transcriptome using StringTie v1.3.3b [24]. Then, filtered reads were aligned to the G. biloba genome. An index of the reference genome was built, and clean paired-end reads were aligned to the reference genome using Hisat2 v2.0.5. Gene expression levels were quantified based on the fragments per kilobase of transcript per million mapped reads (FPKM). Differentially expressed genes (DEGs) between OL and SL were identified using the DESeq2 R package, with padj < 0.05. Kyoto Encyclopedia of Genes and Genomes (KEGG) enrichment analyses of the DEG datasets were performed using the R package cluster Profiler. Sequenced data were deposited in the Genome Sequence Archive (GSA) database under accession number CRA004963.

### 2.5. Quantitative Real-Time PCR

Six genes (*Gb_14030*, *Gb_14029*, *Gb_02188*, *Gb_01519*, *Gb_08543*, *Gb_10073*) associated with flavonoid and terpene lactone biosyntheses were selected and analyzed in OL and SL samples using quantitative real-time PCR (qRT-PCR). Specific primers were designed using Primer Premier 5 software (Premier Biosoft, Palo Alto, CA, USA) (Table S1). *GAPDH* expression in *G. biloba* was used as the internal control, and assays for each gene were conducted in triplicate. Expression levels were quantified using the $2^{-\Delta\Delta Ct}$ method. All reactions were conducted using three biological replicates.

### 2.6. Statistical Analyses

All phenotypic, physiological, and qRT-PCR data are presented as the mean ± standard deviation (SD) of at least three biological replicates. The significance of differences between the OL and SL groups for all experiments was assessed using two-sided Student's t-test and analysis of variance (ANOVA) followed by post hoc tests. *p*-values < 0.05 and <0.01 were considered to indicate significant and highly significant differences, respectively.

## 3. Results

### 3.1. Differences in Morphology between Leaves from Resprouting Stems and Leaves from Old Branches

The morphology of SL was significantly different from that of OL (Figure 1A–F). SL were thicker than OL, with a mean thickness of 3.77 ± 0.66 versus 1.35 ± 0.35 mm, representing an increase of 1.79 times (Figure 1G). Notably, the number of leaf lobes was 8.17 times that of OL (Figure 1H), whereas the leaf area did not differ significantly between SL and OL (Figure 1I). Furthermore, the leaf fresh weight was elevated by 0.67 times, with a mean fresh weight of 7.21 ± 0.53 g for SL versus 4.33 ± 0.22 g for OL (Figure 1J). Similarly, compared with OL, the leaf water content was elevated by 0.19 times in SL (Figure 1K). We further compared the dry weights of OL and SL, but no significant difference was found (Figure 1L).

### 3.2. Transcriptome Analysis of Leaves from Resprouters and Old Branches

SL and OL were collected for transcriptome analysis. The RNA-seq analysis generated a total of 488,907,490 clean reads, and 19,239 and 18,743 genes were identified as expressed in SL and OL, respectively (Table S2, Figure 2A). In total, 2959 DEGs were identified, among which 1160 were upregulated and 1799 downregulated in SL relative to OL (Figure 2B,C). We further conducted KEGG enrichment and Gene Ontology (GO) classification analyses of the DEGs to identify the main pathways associated with the DEGs in SL and OL. GO analysis showed that the DEGs were mainly enriched in the following attributes: photosystem, transferase activity, cellular polysaccharide metabolic process, glucan metabolic process, and carbohydrate-binding (Figure S1). Gene annotations were also searched for in the KEGG database. KEGG enrichment analysis revealed that 114 pathways were enriched among these DEGs, including those associated with phenylpropanoid biosynthesis, flavone and flavanol biosynthesis, metabolic process, and photosystem (Figure S1, Figure 2D).

### 3.3. Changes in the Expression of Genes Associated with Photosynthesis

We identified some genes involved in light responses and the Calvin cycle in photosynthesis. Compared with OL, the expression of eight DEGs related to photosystem II (PS II) (*Gb_22296*, *Gb_18604*, *Gb_22672*, *Gb_09035*, *Gb_03955*, *Gb_ 19153*, *Gb_06826*, and *Gb_26289*) was upregulated in SL. We also identified three DEGs (*Gb_29773*, *Gb_21299*, and *Gb_21286*) related to photosystem I (PS I) that were upregulated in SL. In addition, the expression of *Gb_39846* encoding ATP synthase was upregulated in SL. On the other hand, the expression of *Gb_18205*, encoding ATP synthase, and *Gb_24740*, encoding ferredoxin, was downregulated in SL (Figure 3A).

By contrast, there were few DEGs related to the Calvin cycle. The expression of *Gb_29436*, which encodes fructose-bisphosphate aldolase, was downregulated, whereas that of *Gb_16477*, which also encodes rubisco, and *Gb_14561*, which encodes phosphoglycerate kinase, was upregulated (Figure 3B).

### 3.4. Changes in the Expression of Genes Associated with Plant Hormones

Plant hormones are chemical messengers that regulate numerous physiological processes. Growth-related hormones include gibberellin (GA) and indole-3-acetic acid (IAA). We identified DEGs involved in GA and IAA biosynthesis as well as signal transduction in the OL and SL (Figure 4A–C). Compared with OL, the expression levels of *Gb_38576* and *Gb_31184* encoding ent-kaurene oxidase, which is related to GA synthesis, were increased in SL. Conversely, the expression levels of *Gb_06342*, encoding ent-kaurenoic acid oxidase, and *Gb_19227*, encoding gibberellin 20 oxidase, were significantly decreased in SL (Figure 4A). Similarly, in terms of GA signaling, the expression levels of genes encoding DELLA (*Gb_34637* and *Gb_22850*), a repressor of the GA signaling pathway, was increased significantly, whereas the expression levels of genes encoding positive receptors involved in GA signaling, i.e., *Gb_17753*, which encodes the gibberellin receptor (*GID1*), and *Gb_11704*, which encodes an F-box protein (*GID2*), were significantly decreased in

SL (Figure 4B). The expression levels of three genes involved in IAA signal transduction, *Gb_25859*, *Gb_14852*, and *Gb_ 31741*, which encode the auxin transporter, were significantly increased (Figure 4C).

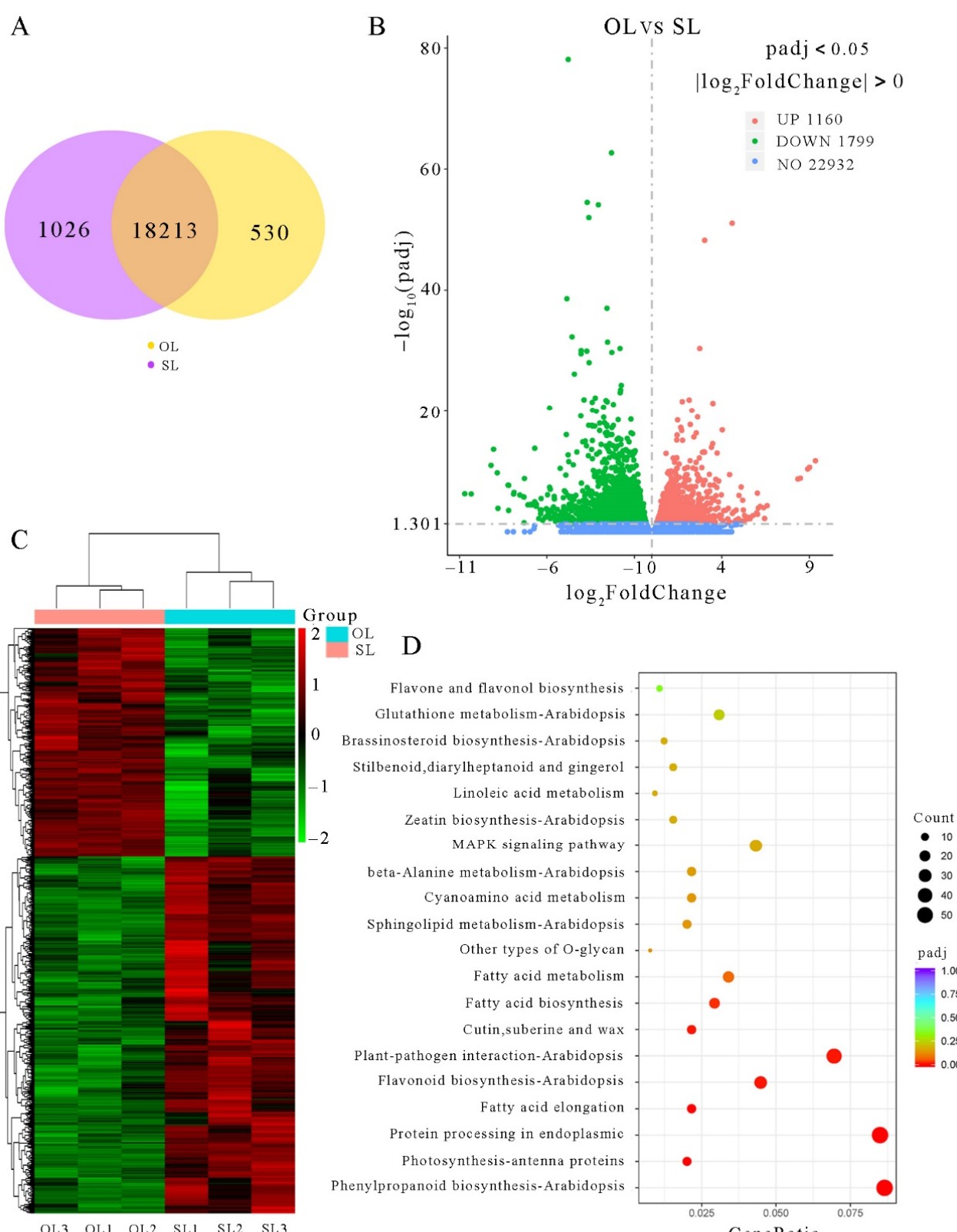

**Figure 2.** The DEGs and results of enrichment analysis: (**A**) Venn diagram of the DEGs; (**B**) Volcano plot of the DEGs; (**C**) Cluster plot and heat map of the DEGs; (**D**) Results of KEGG enrichment analysis.

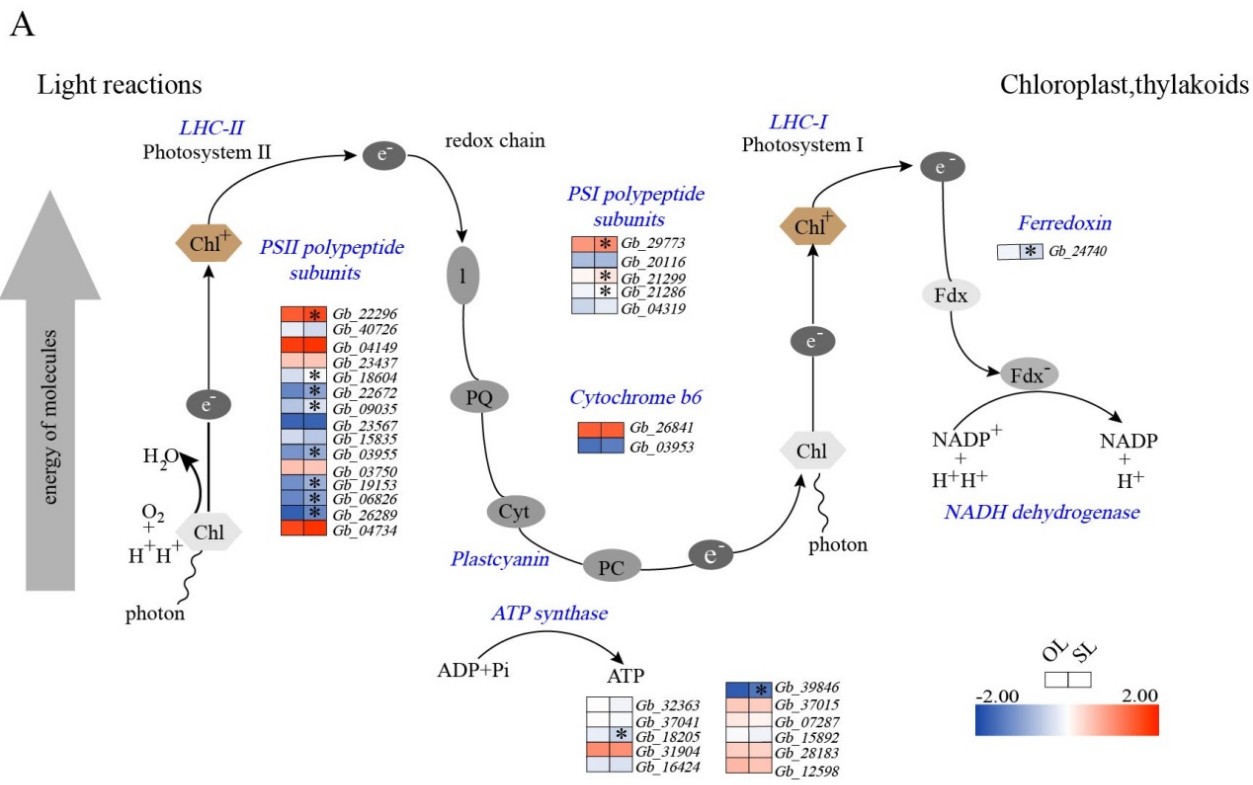

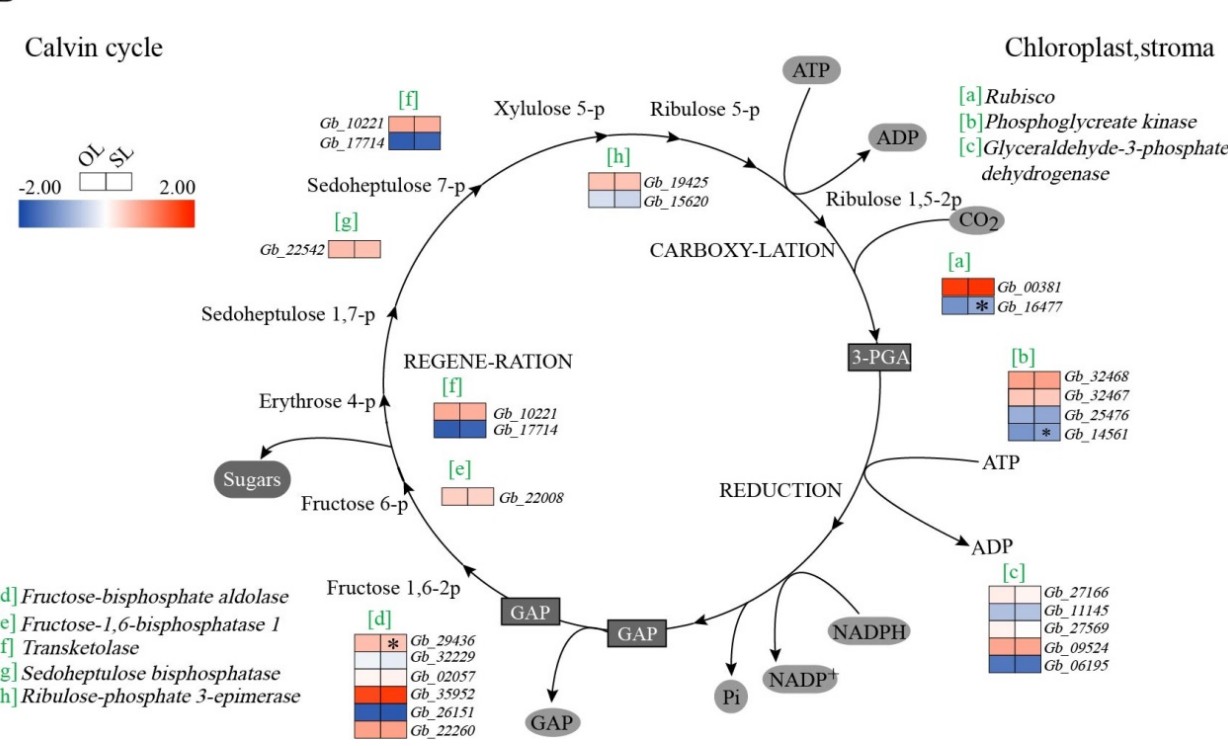

**Figure 3.** Changes in the expression of photosynthesis-related genes: Pathway diagrams of (**A**) light reactions; (**B**) the Calvin cycle in photosynthesis, with color-coded squares indicating changes in gene expression. Pathways were drawn based on information from the literature and the KEGG database. * $p < 0.05$ compared with OL.

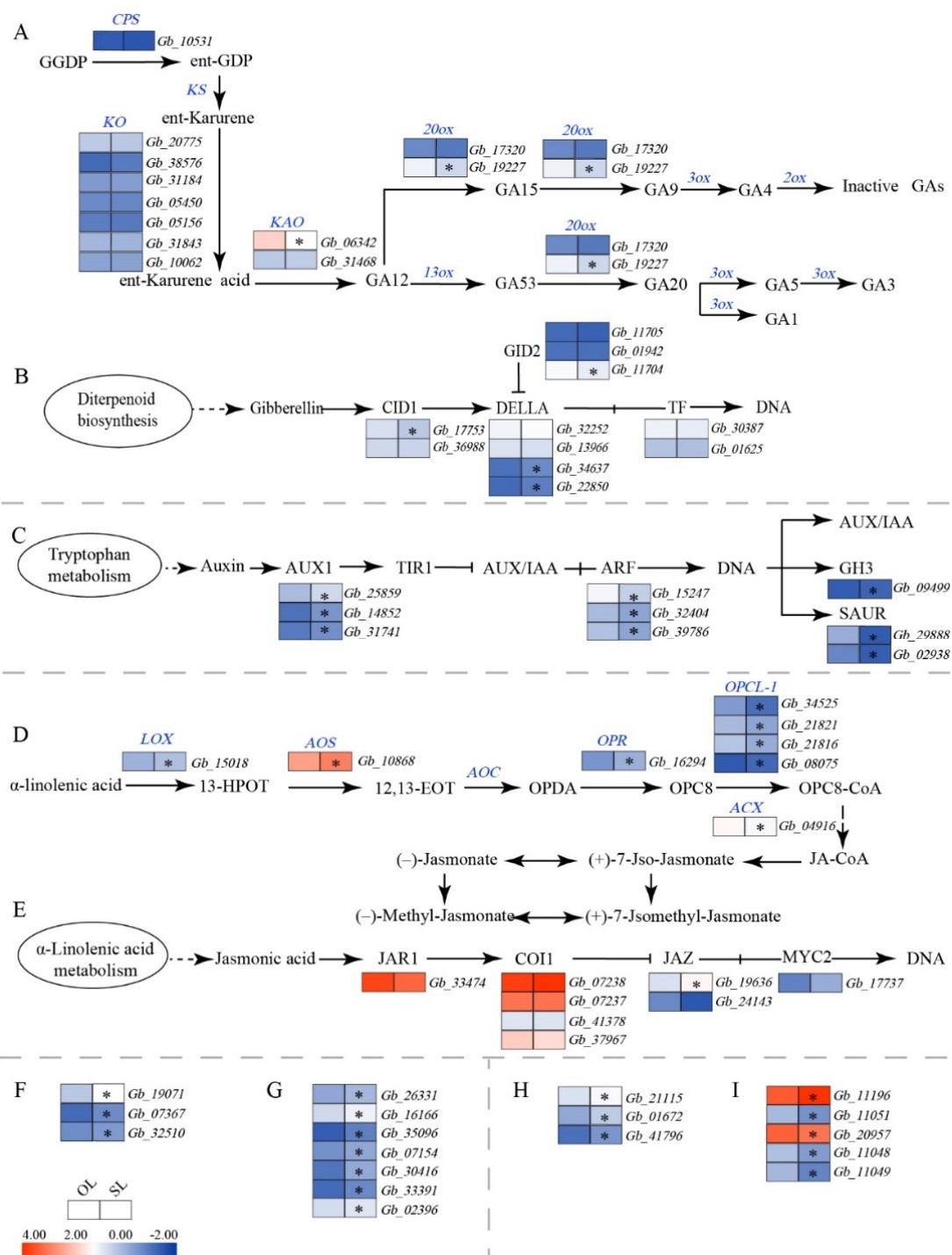

**Figure 4.** Changes in the expression of hormone-related genes: (**A**) Expression of DEGs in the GA biosynthesis pathway; (**B**) Expression of DEGs in the GA signaling pathway; (**C**) Expression of DEGs in the IAA signaling pathway; (**D**) Expression of DEGs in the JA biosynthesis pathway; (**E**) Expression of DEGs in the JA signaling pathway; (**F**) Expression of DEGs in the ABA biosynthesis pathway; (**G**) Expression of DEGs in the ABA signaling pathway; (**H**) Expression of DEGs in the SA biosynthesis pathway; (**I**) Expression of DEGs in the SA signaling pathway. Pathways were drawn based on information from the literature and the KEGG database. * $p < 0.05$ compared with OL.

We also identified DEGs involved in the biosynthesis and signal transduction of stress-associated plant hormones, such as jasmonic acid (JA), abscisic acid (ABA), and salicylic acid (SA). Regarding genes involved in the JA synthesis pathway, the expression levels of *Gb_15018* encoding linoleate 9S-lipoxygenase, *Gb_10868* encoding allene oxide synthase,

*Gb_16294* encoding 12-oxophytodienoate reductase, and *Gb_08075* encoding 4-coumarate: coenzyme A (*CoA*) ligase (*OPCL-1*) were increased in SL. By contrast, the expression levels of *Gb_04916* encoding acyl-CoA oxidase and *Gb_34525*, *Gb_21821*, and *Gb_21816* encoding *OPCL-1* were markedly decreased in SL (Figure 4D). For genes involved in the JA signal transduction pathway, compared with OL, the expression level of *Gb_19636* encoding the protein TIFY 3B (*JAZ*) was upregulated in SL (Figure 4E). For genes involved in the ABA synthesis and signaling pathways, the expression levels of *Gb_19071* and *Gb_07367*, which encode 9-cis-epoxycarotenoid dioxygenase, and *Gb_32510*, which encodes zeaxanthin epoxidase, were notably higher in SL (Figure 4F). The expression levels of genes encoding the ABA receptor (*PYL*; *Gb_26331*, *Gb_16166*, and *Gb_35096*), protein phosphatase 2C (*PP2C*; *Gb_07154*, *Gb_30416*, and *Gb_33391*), and serine-threonine protein kinase (*SnRK2*; *Gb_02396*) were markedly increased in SL (Figure 4G). We also observed a prominent increase in the expression levels of *Gb_21115*, *Gb_01672*, and *Gb_4179*, which encode phenylalanine ammonia-lyase (*PAL*) and are associated with SA synthesis, in SL (Figure 4H). In addition, the expression of *Gb_11196*, which encodes pathogenesis-related protein 1 and is related to SA signal transduction was also significantly upregulated in SL (Figure 4I).

*3.5. Expression of Flavonoid Biosynthesis-Related Genes in Leaves from Resprouters and Old Branches*

As flavonol is the main active component of ginkgo leaves, we compared the levels of flavonol components between OL and SL. The total flavonol glycoside content was significantly higher in SL (by 18.9–31.3%; Figure 5A), with $7.62 \pm 0.32$ mg/g in SL compared with a content of $6.04 \pm 0.70$ mg/g in OL. In particular, the kaempferol and isorhamnetin contents in SL were higher by 129.2% and 61.3%, respectively, although there was no significant difference in the quercetin content (Figure 5B). We further examined the expression of genes involved in the flavonoid biosynthesis pathway. Through transcriptome data analysis, we identified key structural genes upregulated in SL encoding *PAL* (*Gb_41796*, *Gb_21115*, and *Gb_01672*), cinnamate-4-hydroxylase (*C4H*; *Gb_40727*), 4-coumarate: CoA ligase (*4CL*; *Gb_21816*, *Gb_21821*, and *Gb_34525*), flavonol synthase (*FLS*; *Gb_14030* and *Gb_14029*), dihydroflavonol-4-reductase (*DFR*; *Gb_26459*), anthocyanidin synthase (*ANS*; *Gb_21870*), and anthocyanidin reductase (*LAR*; *Gb_09086*, *Gb_09087*, and *Gb_10028*). By contrast, the genes encoding chalcone synthase (*CHS*; (*Gb_01519*) and flavonoid 3′-hydroxylase (*F3′H*; *Gb_02188*) were downregulated (Figure 5C). Next, using qRT-PCR, we analyzed the expression profiles of *FLS* (*Gb_14030* and *Gb_14029*) (Figure 5D,E), *F3′H* (*Gb_02188*) (Figure 5F), and *CHS* (*Gb_01519*) (Figure 5G), which encode key enzymes involved in flavonoid biosynthesis. The *FLS* expression level was markedly higher in SL than in OL. The qRT-PCR results for *FLS*, *CHS*, and *F3′H* were in accordance with the transcriptome data.

*3.6. Identification of Terpene Lactones in Leaves from Resprouters and Old Branches*

Terpene lactones were reported as key active ingredients in *G. biloba* leaves. We measured the terpene lactone content and found that the total terpene lactone content tended to be higher in SL than in OL (Figure 6A). Although slightly less bilobalide was found in SL, the ginkgolide A content was 2.04-fold higher than that in OL, representing the largest difference in content among all terpenoid lactones. The ginkgolide B content in SL was 1.45-fold higher than that in OL (Figure 6B). We further examined the expression of genes involved in the terpene endolipid biosynthesis pathway. Only key structural genes encoding geranylgeranyl pyrophosphate synthase (*GGPS*; *Gb_08543* and *Gb_10073*) and involved in the methylerythritol phosphate (MEP) pathway were found to be upregulated in SL. By contrast, key structural genes encoding acetyl-CoA acetyltransferase (*Gb_04657*) and 3-hydroxy-3-methylglutaryl-CoA reductase (*Gb_20551*, *Gb_26605*, and *Gb_37788*) and involved in the mevalonate pathway were downregulated (Figure 6C). We further analyzed the expression profile of *GGPS* using qRT-PCR. Although the expression level of *Gb_08543* was also markedly higher in SL, the qRT-PCR result for *Gb_10073* was not in accordance with the transcriptome analysis result, which showed that *Gb_10073* was downregulated (Figure 6D,E).

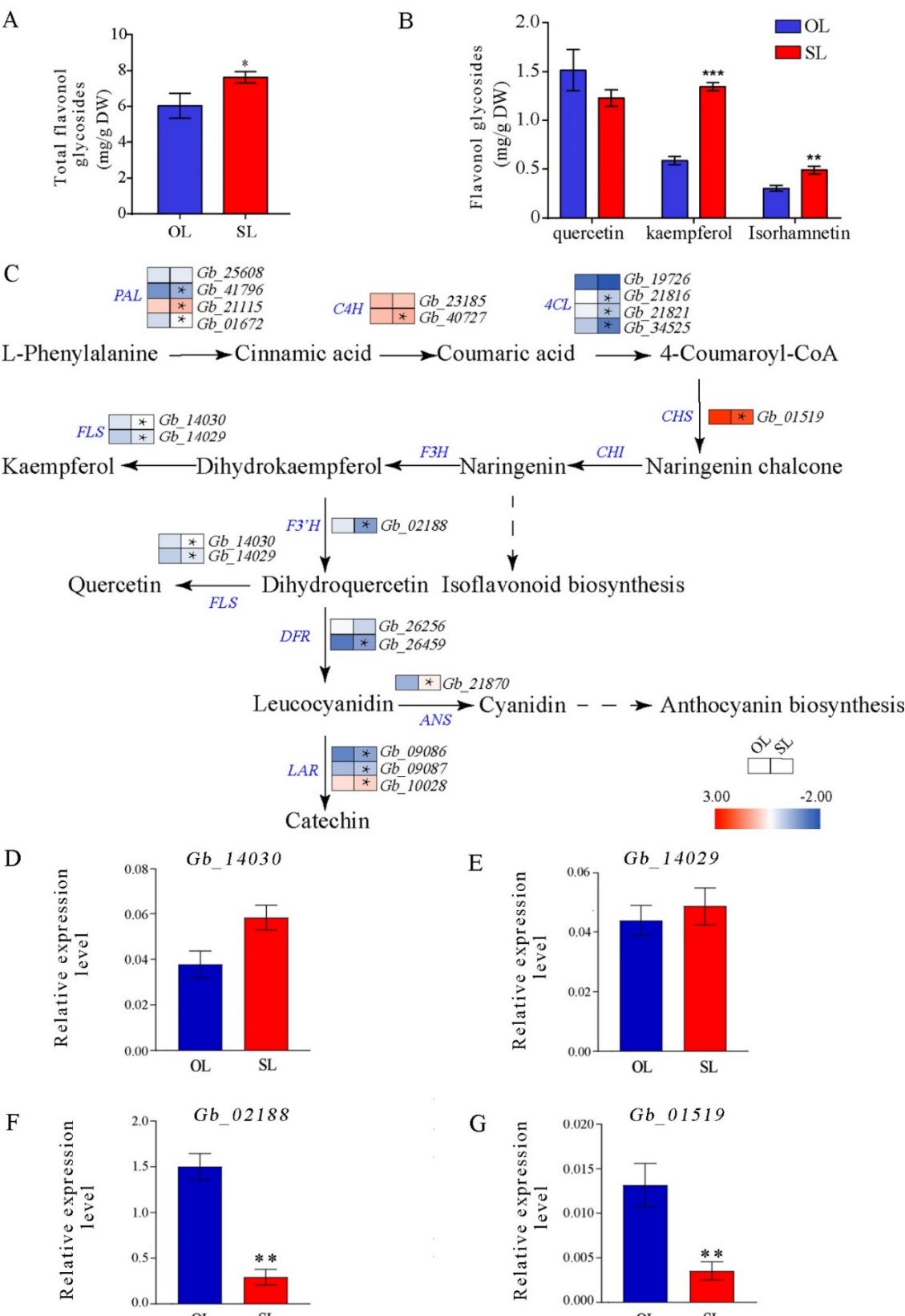

**Figure 5.** (**A**) Total flavonol glycoside contents; (**B**) Individual flavonol glycoside contents; (**C**) Flavonoid biosynthesis pathway; (**D**) Relative expression levels of *Gb_14030* (*FLS*); (**E**) Relative expression levels of *Gb_14029* (*FLS*); (**F**) Relative expression levels of *Gb_02188* (*F3′H*); (**G**) Relative expression levels of *Gb_01519* (*CHS*). $*$ $p < 0.05$, $**$ $p < 0.01$, $***$ $p < 0.001$ compared with OL.

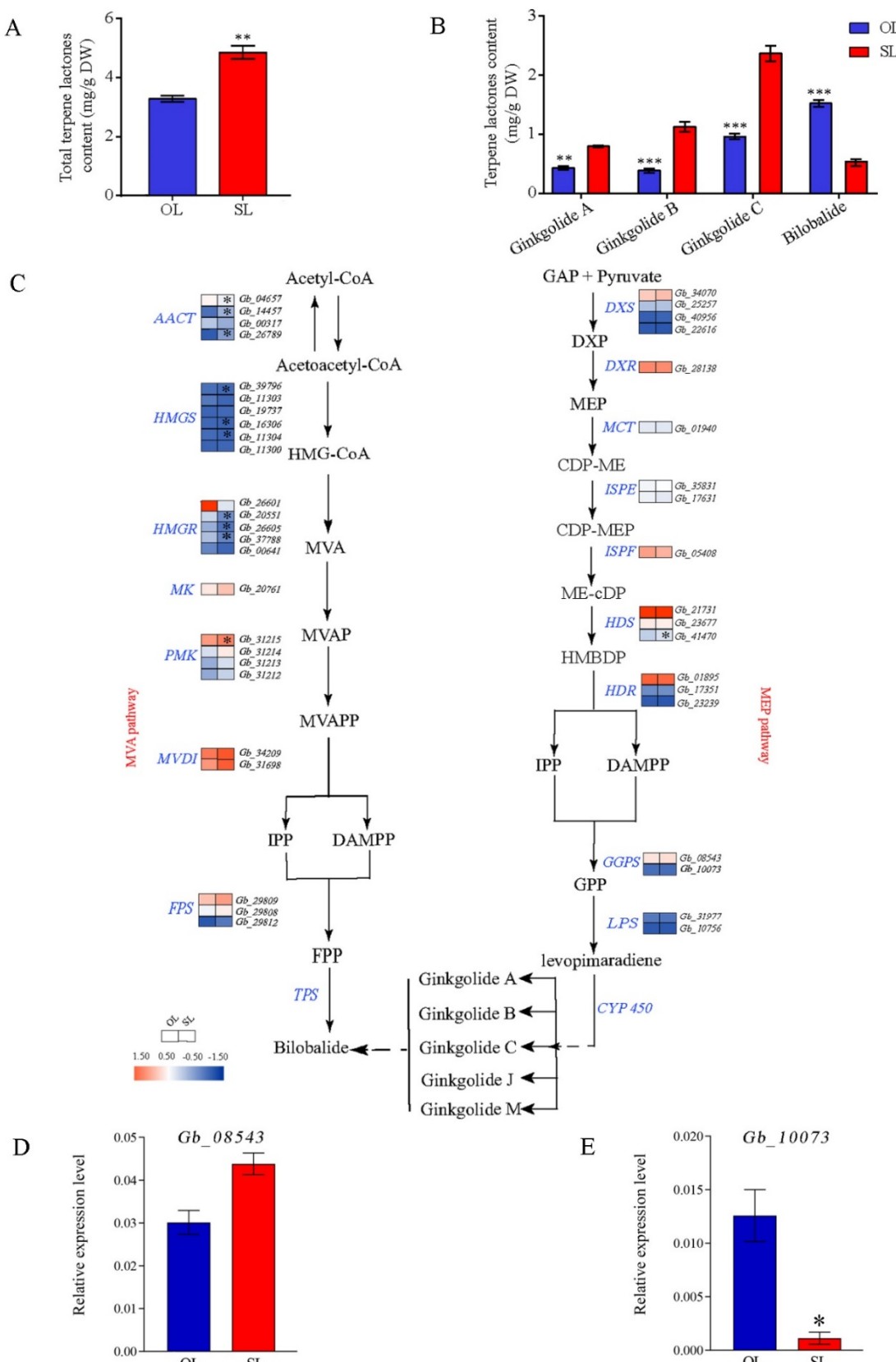

**Figure 6.** (**A**) Total terpene lactone contents; (**B**) Individual terpene lactone contents; (**C**) Terpenoids biosynthesis pathway; (**D**) Relative expression levels of *Gb_08543* (*GGPS*); (**E**) Relative expression levels of *Gb_10073* (*GGPS*). * $p < 0.05$, ** $p < 0.01$, *** $p < 0.001$ compared with OL.

## 4. Discussion

### 4.1. Resprouting and Rejuvenation in Old Ginkgo Trees

*Ginkgo biloba* is a well-known long-lived tree species; its life can reach hundreds of years or even thousands of years [16]. Different from some old trees, the number of new stems often resprout at the base of the trunk in old ginkgo trees. For example, an old ginkgo tree, in the Tianmu Mountains in Zhejiang Province of China, its original trunk had been destroyed, whereas the existing five big trunks are all developed from resprouters [17]. In addition, some old ginkgo trees that survived from resprouting can be found in China, such as Yunnan, Guizhou and Sichuan. From these phenomena, it is possible that resprouting is largely related to the survival strategy of the old ginkgo tree. In our study, we investigated a 544-year-old female ginkgo tree and found that resprouting occurred along the annular strip at the outer edges of the trunk base. The resprouting stems grew vigorously and formed a cluster state. Because resprouting is a response to damage or senescence [25–27], the numerous stems that had resprouted from the base of this old tree are likely an indication of the resilience of *G. biloba*.

Resprouting could reverse the developmental stage of adult or aging plants to restore their chronological age to juvenile status and to regain the developmental potential of young individuals [28]. Generally, the stems of resprouters exhibit characteristics of rejuvenation and differ in morphology and from that of adult or old trees. Differences in foliar and stem morphology between juvenile and adult plants are often the most significant features of developmental phase changes [29]. For example, the juvenile leaves of *Eucalyptus globulus* are wider, larger, and have a greater fresh mass, palisade cell length, and stomatal density compared with mature leaves [30]. Previous studies also revealed that in ginkgo, the leaves of young seedlings are larger with more leaf lobes, but as the tree ages, the leaf area becomes smaller and the number of leaf lobes decreases significantly [22]. In our study, we found that the leaves from resprouting stems had considerably more leaf lobes. Additionally, compared to the leaves from old branches, the thickness and water content of the leaves from resprouters were substantially higher. Considering leaf lobes are the important juvenile characteristics in *G. biloba*, our results suggested that resprouting stems are still in the juvenile state in old ginkgo trees.

### 4.2. Photosynthesis Associated Gene Expression

Leaves are the most important photosynthetic organs of plants, which photosynthesis and transpiration are the basis of plant growth and development [31]. Some studies have found that leaf net photosynthetic activity increased in resprouters or rejuvenated plants in many species, and renew branches have the increased growth amount, leaf water potential and non-structural carbohydrate of leaves [32]. In *Sequoia sempervirens*, compared with adult shoots, juvenile and rejuvenated shoots have higher photosynthetic rates and chlorophyll contents, but as plants age and grow, the rate of photosynthesis often decreases [33].

In our previous studies, through analyzing tree rings, we selected six old ginkgo trees from 193-year-old to 667-year-old and found that, compared with mature trees, leaf areas and photosynthetic efficiencies did not change significantly in these old trees, suggesting that hundred years old ginkgo trees are still in a healthy and mature state [16]. Here, through transcriptome analyses, we found that compared with leaves from branches, some PSII- and PSI-related protein subunits were upregulated in SL. PSII is vital for the initiation of photosynthesis and the promotion of electron transport [34,35]. PSI has a critical role in the metabolic networks and physiological responses in plants [36]. PSI contents in higher plants appear to be highly constant, especially once the photosynthetic apparatus has been fully established, such as, in mature leaves [37]. In our study, some DEGs in PSII, PSI and the gene encodes rubisco (*Gb_16477*) were upregulated in the leaves of resprouters than those of old branches, suggesting that the photosynthesis of leaves from resprouting may be more active than that of old branches.

### 4.3. Plant Hormone-Associated Gene Expression

Plant hormones are a chemical messenger produced at a very low concentration in regulating the growth and development of a variety of plants [38]. As an important plant endogenous hormone, GA and IAA are almost involved in every process of development of higher plants. Some studies found that GA plays an important role in plant rejuvenation [3]. However, in this study, we found that the expression levels of *Gb_17753* (*GID1*) and *Gb_11704* (GID2) were significantly lower in SL, but the suppressor proteins DELLA expression increased, indicating that GA signaling pathway transduction may be inhibated in resprouters. Differently, the expression of some genes encoding AUX, such as *Gb_25859*, *Gb_14852*, and *Gb_31741* were significantly increased in SL, suggesting that IAA may take a pivotal role in the growth of leaves in the resprouting stems in old ginkgo trees.

There are also some stress-related plant hormones, such as JA, ABA and SA, which play critical roles in helping the plants to adapt to adverse environmental conditions [39]. For example, under osmotic stress, the expression level of several ABA biosynthesis genes (such as *ZEP*, *NCED*) is upregulated [40]. In addition, JA and SA are involved in the defense response to biotic stress [41,42]. Under abiotic stress, the expression of *LOX* in the JA synthesis pathway is induced [42]. To limit the growth of pathogens and their access to water and nutrients in plants, the activation of SA-dependent signaling pathways could lead to the expression of *PRP* [43]. In this study, we found that the expression levels of *Gb_19071* (*NCED*), *Gb_32510* (*ZEP*), *Gb_15018* (*LOX*) and *Gb_11196* (*PRP*) in SL all increased significantly. Thus, we speculate that these stress-responsive genes may enhance the defensive capacity in the leaves of resprouters in old ginkgo trees.

### 4.4. Secondary Metabolite Accumulation

To cope with constraints due to stress from growing under challenging and changing conditions, plants usually produce metabolites [21]. At present, a number of secondary metabolites including terpene trilactones, flavonoid glycosides, and lignin were isolated from plants [44]. These compounds play antibacterial, antioxidative, and antimicrobial roles [45]. Although tree developmental age, soil type, and natural variations resulting in plant genetic diversity all contribute to the wide variation in terpene content [46], tree age is the main factor influencing terpene content; compared with the leaves of young trees, the terpene content in the leaves of old trees is usually lower.

In *G. biloba*, flavonoids and terpenoid lactone are the main secondary metabolites. They are also active components of *G. biloba* extracts used for treating cardiovascular diseases, dementia, and dizziness [47,48]. The total terpene lactone and flavonoid glycoside contents were found to be higher in the leaves of young ginkgo trees than in those of old trees [22]. In our study, the total flavanol and flavanol glycoside contents were significantly higher in the leaves from SL. Furthermore, the total terpene lactone and ginkgolide A, B, and C contents tended to be higher in SL than in OL. These results indicate that the production of flavonoids and terpenoid lactones may be increased in resprouters from old *G. biloba* trees.

We further identified DEGs between SL and OL involved in phenylpropanoid, flavone, and flavonoid biosynthesis pathways. The flavonoid biosynthesis pathway is extremely complicated, as its metabolic processes are directly affected by many genes encoding key enzymes [23]. In our study, the expression of 14 genes associated with the flavonoid biosynthesis pathway, i.e., those encoding *PAL* (*Gb_41796*, *Gb_21115*, and *Gb_01672*), *C4H* (*Gb_40727*), *4CL* (*Gb_21816*, *Gb_21821*, and *Gb_34525*), *FLS* (*Gb_14030* and *Gb_14029*), *DFR* (*Gb_26459*), *ANS* (*Gb_21870*), and *LAR* (*Gb_09086*, *Gb_09087*, and *Gb_10028*), was upregulated in the leaves of basal sprouts. Regarding terpenoid lactone biosynthesis, only *GGPS* genes (*Gb_08543* and *Gb_10073*), which are involved in the MEP pathway, were upregulated in SL. These results indicate that secondary metabolite production could effectively be activated in resprouters arising from old ginkgo trees.

## 5. Conclusions

In this study, we compared the OL and SL of a 544-year-old ginkgo tree by analyzing morphological changes, physiological indices, and RNA-seq data. SL were thicker with more leaf lobes, a higher water content, and exhibited juvenile phenotypic characteristics. Some genes involved in PSII and PSI were upregulated in SL. Additionally, the expression of genes related to the auxin signaling pathway and stress-related hormones was also upregulated in SL. These results imply that changes in hormone metabolism, especially auxin metabolism, may be an important factor for leaf juvenility. Interestingly, the contents of the most important secondary metabolites, kaempferol, isorhamnetin, ginkgolide A, ginkgolide B, and ginkgolide C, improved significantly. The increased accumulation of flavonoids and terpene lactones in leaves of resprouters may be regulated by *PAL*, *FLS* and *GGPS*.

**Supplementary Materials:** The following are available online at https://www.mdpi.com/article/10.3390/f12091255/s1, Table S1: Primer sequences used in this study, Table S2: RNA-Seq data information, Figure S1: Results of Go enrichment analysis.

**Author Contributions:** W.L. and L.W. conceived and designed the experiments. J.Y., S.Z. and M.T. wrote the paper, J.Y., S.Z., M.T., J.L., T.W. and Y.X. performed the experimental work and contributed the data analyses. J.Y. and S.Z. contributed equally to this work. All authors have read and agreed to the published version of the manuscript.

**Funding:** This research was funded by the National Natural Science Foundation of China (No.793 31971686, 31670695), and the Undergraduate Student Scientific Research Innovation Projects in Jiangsu Province (202011117120Y).

**Institutional Review Board Statement:** Not applicable.

**Informed Consent Statement:** Not applicable.

**Data Availability Statement:** The raw sequence data reported in this study have been deposited in the Genome Sequence Archive (GSA) database under the accession number CRA004963.

**Acknowledgments:** We thank Novogene Bioinformatic Technology for helping with sRNA-sequencing and technical assistance. This project was supported by the College of Student Innovation and Entrepreneurship Training Projects Jiangsu Province (Grant No. 202011117120Y). We are grateful to Ningtao Xu, Xinyu Mao and Xiaoyin Gu for the help with sampling and experiments.

**Conflicts of Interest:** The authors declare no conflict of interest.

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
