# Peer review of "Physiological and Genetic Analysis of Leaves from the Resprouters of an Old Ginkgo biloba Tree"

_forests, doi:10.3390/f12091255_

Round 1

Reviewer 1 Report

Broad comments.

The ginkgo tree is very large and old (Figure 1A) and it should be detailed which old branches (L.57) were used to collect the leaves. Leaves from the top/bottom, north/south, inner/outer, etc. parts of the crown can be very different from each other. First, the uniformity of the leaves within the crown had to be confirmed, and then the crown leaves had to be compared with the leaves of tillers. 10-15 leaves from a large tree is clearly not enough for correct statistical analysis.

It is incorrect to write about changes in traits at the whole plant level only on the basis of increased gene expression:

L.172-173, 305-306. Increased expression of Rubisco and phosphoglycerate kinase genes does not necessarily enhance photosynthetic capacity, which the authors did not measure.

L.337-338. The authors did not measure hormone levels, but only gene expression was assessed.

L.218-220, 340-341. Upregulation of the expression of only one pathogenesis-related protein gene does not mean an increased pathogen resistance of the whole plant.

Specific comments.

Subsection 2.3. The description of the method is incomplete. The standard analysis of total flavonoids includes the use of AlCl3, NaNO2 and NaOH, which are not included in the above procedure. The complete protocol should be given and the flavonoid used for calibration should be indicated.

Subsection 2.6. Genes that were analyzed by qRT-PCR should be indicated.

Subsection 2.7. “All phenotypic, physiological, and qRT-PCR data are presented as the mean ± standard deviation (SD) of at least three biological replicates”, but Fig. 1 “Error bars represent the mean ± SD (n = 2)”. Clarify please.

L. 126-128. This text is taken from the template file and should be deleted.

Fig. 1K. According to the figure, the water content in the leaves of old branches is about 25%. This is less than the water content in wood! In addition, according to the text, the water content in the leaves of tillers is 97.1% higher (L. 136-137), while in the figure they differ by about 3 times (i.e. 200%). Correct please.

Fig. 5, 6. For Fig. 5A,B and 6A,B the type of statistical analysis should be indicated.

L. 268-292. This material should be moved from the Discussion to the Introduction, which is very poor.

Reviewer 2 Report

Dear Authors,

I read your manuscript with great interest. You have done a lot of experimental work. Your comprehensive approach is undoubtedly a plus. However, I must inform you that the manuscript requires a major revision for publication. Please read my comments:

The "Abstract" part needs some improvement. Ideally, this part should include a summary of all parts of the article: introduction, materials and methods, results and discussion, conclusions. I recommend that you add information on the biological and economic value of ginkgo. It is necessary to mention the rejuvenization of old trees, to describe in more detail the sprout tillers. Next, you should briefly formulate the purpose of your work and the tasks that need to be completed. The description of the results can be left as is. The conclusions at the end of "Abstract" should make it clear whether you have achieved your goal or not.
Line 13. Please explain the abbreviation "SL". Why do you write "SL" and not "ST", abbreviating "sprout tillers"? Perhaps you should replace this abbreviation with a more accurate one throughout the manuscript.
Lines 13, 50 and 58. You are writing about a 544 year old ginkgo tree. How did you determine the exact age of such an old tree without sawing off and counting annual rings? If the age is established from the surviving documents, then they must be mentioned in the manuscript.
Line 24. Please replace "RNA-seq" with "RNA sequencing".
The "Introduction" part is extremely small. It needs major improvement. More information on sprouting needs to be added. I advise you to use the text 268-326 lines (I think it is inappropriate to write about this in the "Discussion"). You should rework it for "Introduction".
Line 48. This is where the purpose of your work should be clearly stated.
Line 55-124. The methods were described in sufficient detail. I see that this part of manuscript is good.
Lines 126-128. Delete this text. It has nothing to do with your work.
Lines 130-138. I find it extremely unfortunate to express the difference as a percentage in comparison. Better to use "1.8 times more", etc.
Figures 3, 4, 5, 6. These figures are of the same type and very cumbersome. Please consider the proposal to keep only one of them in the manuscript (for example, 3) and to transfer the others to the supplementary materials.
Lines 162, 169. You write "calvin cycle" and "Calvin cycle". Pick one variant and use it throughout the manuscript.
Part "Discussion". This part is extremely poorly written. In fact, there is no discussion in the text. Text in 268-326 lines is not a discussion. This text needs to be revised and moved to the "Introduction" part. Please carrifully re-write the "Discussion" part. I recommend that you take each result from the Results section as a fulcrum. It is necessary to carry out serious work with the literature on each of your results and each of your conclusions. Compare your results and conclusions with those of colleagues who have already published their articles. Try to avoid speculation and expressions like "higly likely" in the text. This is unacceptable for a serious scientific discussion.

Letters and inscriptions in all figures need to be made clearer.

Your supplementary materials are not available to me for viewing. Make sure you download them.

Round 2

Reviewer 1 Report

The manuscript has been improved and can be accepted for publication in this form.

Author Response

Thank you for your recognition of our revised manuscript

Reviewer 2 Report

Lines 68-71. I have to repeat point 6. I asked you about the purpose of your work. Your answer did not satisfy me. You do not see the difference between the goal of the work and its results. The purpose of the work is what you strive for. The results of the work are what you got in the course of the work. Write the PURPOSE of your work, please.

Lines 465-466. What is “20-, 200-, and 600- y-old trees”? Please rewrite sentences, as in line 101, for example.
